# Gender differences in Dutch research funding over time: A statistical investigation of the innovation scheme 2012–2021

**Casper Albers** [1] *, **Sense Jan van der Molen** [2], **Thijs Bol** [3]

**1** Heymans Institute for Psychological Research, University of Groningen, Groningen, The Netherlands, **2** Institute of Physicics, Leiden University, Leiden, The Netherlands, **3** Department of Sociology, University of Amsterdam, Amsterdam, The Netherlands

\* c.j.albers@rug.nl

## Abstract

### Background

In 2015, the Dutch research council, NWO, took measures to combat gender bias disadvantaging female applicants in a popular three-tiered funding scheme called the Talent Programme. The innovation scheme consists of three grants for different career stages, called Veni, Vidi and Vici.

### Objectives

This paper studies the question whether or not NWO has been successful in removing gender differences in their funding procedure.

### Methods

Using all available data from 2012 onwards of grant applications in the Talent Programme (16,249 applications of which 2,449 received funding), we study whether these measures had an effect using binomial generalized linear models.

### Results

We find strong statistical evidence of a shift in gender effects in favour of female applicants in the first tier, the Veni ($p < .001$). Significant gender differences are not found in the two other tiers, the Vidi and Vici schemes.

### Conclusions

In recent years, female applicants are more likely to be awarded with a Veni grant than male applicants and this gender gap has increased over time. This suggests that gender differences still exist in the assessment of Talent Programme submissions, albeit in a different direction than a decade ago.

**Data Availability Statement:** All data and R scripts are available from https://osf.io/8bfaz/.

**Funding:** The author(s) received no specific funding for this work.

**Competing interests:** The authors have declared that no competing interests exist.

## Introduction

Do male and female academics have equal opportunities in obtaining research grants? While some existing studies that have addressed this question found that men were favored when applying for grants [1], the vast majority found equal funding rates across the two genders [2–9]. In this study we take a temporal perspective, and investigate whether and how this gender disbalance has changed over time.

We study the case of the Netherlands, where a key source of research funding is the Talent Programme (TP) of the Dutch Research Council, NWO. Grants from this funding scheme are personal, and are allocated based on the quality of the researcher, quality of the proposal, and potential for knowledge utilization. The TP consists of three tiers called Veni, Vidi, and Vici, respectively, after Julius Caesar's (in)famous phrase. Veni-grants (at most 280k€) can be applied for by young scientists who are within three years of receiving their PhD-degree. Vidi-grants (at most 800k€) can be applied for by scientists up to eight years after receiving their PhD-degree, and Vici-grants (at most 1.5M€) are open to those within fifteen years of obtaining their PhD-degree. In certain situations, such as childcare responsibilities, these terms can be extended.

The TP is an interesting case to study temporal changes in gender differences in research funding for two reasons. First, the grants are personal, and the (assessed) quality of the applicant plays a big role in whether the grant will be funded or not. Earlier research found that particularly in the assessment of the quality of the researcher, gender plays a role—although that did not lead to eventual differences in research funding [4].

Second, gender inequality in the TP and more particularly the Veni grant has been studied extensively in recent years. In 2015, [10] concluded that this grant scheme disadvantaged women, which led to discussion in the Dutch parliament [11]. Despite methodological criticism [2, 12] on the analyses that formed the basis of these discussions, NWO decided to take several measures to combat gender bias in their funding schemes, such as introducing implicit bias training for committee members. Now that the measures taken by NWO have had considerable time to take effect, we want to describe whether we indeed see changes over time.

While we focus mostly on the Veni scheme for the abovementioned reason, the policy changes were implemented in the Vidi and Vici scheme as well, and these will also be studied. However, due to the relatively small number of applications and grants in these schemes, strong statistical inferences cannot be drawn.

Studying gender inequality more extensively in the early career (for the Veni grant) has another reason too: several studies find that inequalities accumulate over careers [13]. If it is indeed the case that there is gender inequality in research funding in the early career, this will likely have lasting effects on the remainder of the careers of applicants.

To investigate our research question, we apply and compare four possible statistical models, with increasing complexity, for each of the three tiers. For the Veni tier, all models find significant differences between the succes rates of male and female applicants. The models also show a trend over time, where male applicants gradually have lower success rates and female applicants gradually have higher success rates. For the Vidi and Vici tier, no gender differences are found.

## Gender inequality in science funding

Gender inequality in science has been a topic of concern and study for many years [14, 15]. Numerous scientific findings have shed light on the disparities faced by women in scientific fields. For example, when it concerns hiring, some studies find evidence that women have a lower probability to get hired [16]—although there are also studies that find the opposite,

where female scholars obtain tenure earlier in their career [17]. Similarly, some studies find large gender differences in academic (self)-citations [18], whereas others do not [19].

What are the mechanisms that explain these gender disparities? An often heard mechanism is (implicit) bias: under equal performance, people rate the academic performance of women lower than that of men. Studies have revealed the presence of implicit biases that affect the evaluation and perception of women's abilities in scientific settings [20]. These biases can lead to gender-based discrimination in hiring, promotion, and funding decisions, hindering women's career advancement. In many studies, evidence for implicit bias is difficult: often the designs are behavioral, and teasing out the true quality of academic performance is difficult—particularly when using outcomes such as citations that themselves can be affected by gender bias too [21].

The current study is no exception, and we will not be able to know whether the (absence of) observed inequality reflects processes of bias. A particularly difficult concern when it comes to gender inequality in grant funding is a process of self-selection prior to the competition [4, 22]. Obtaining a grant is only possible if one applies for it, and this might affect the gender inequality in grant funding [13]. For example, when male scholars self-select more often into highly competitive grants, their success rate might be lower than women, but this does not necessarily signal that they are discriminated against—or vice versa.

Several studies have investigated (other) aspects of gender bias in Dutch academia; e.g. during the PhD-trajectory, i.e., before being eligible for a Veni-grant [23], or after receiving a grant [24]. A very recent study [4] had an objective similar to ours: to study gender effects in the NWO Talent Programme. In their case, the authors studied confidential assessment reports to find that, in the end, there is no evidence for gender effects in the final funding, although males did receive significantly better reviews. They conclude that juries tend to correct for this gender imbalance when taking the final decision to award grants. Whereas [4] use data up to 2016, we also include more recent data, up to 2022. The main contribution of our study, compared to that of [4], is that we focus on interactions between gender on the one hand and year on the other.

## Change over time

Why do we expect gender inequality in research funding to change over time? Focussing on the temporal variation in gender inequality in grant funding also makes that we do not necessarily have to hypothesize on whether the observed effect is caused by bias or another process: we are mostly interested describing how the inequality changes over time, rather than finding the causal mechanisms for this.

In the Dutch case we have reasons to believe that there are temporal changes. At the start of the TP there was already some focus on gender (inequality) in awarded grants. Panelists, for example, obtained the instruction from NWO to specifically consider female applicants, as a larger NWO goal was to award male and female applicants proportionally: given the amount of applications, men and women should have the same probability to obtain a grant. One practical implication of this policy is that at an ex aequo around the funding threshold, female applicants were prioritized [4]. At the same time, in many aspects there was not much attention for the issue yet. Announcements of the projects that were awarded with a Veni, Vidi, or Vici grants were split up by academic field, but not by gender.

The focus on gender inequality in research funding became stronger in the 2010s, most clearly after a study from 2015 [10] in which the authors found that women were less likely to be awarded with Veni grants and got lower scores for their academic profiles. While part of the methods and results were questioned [2, 12], the media response to the findings was very

strong. One of the largest newspapers in the Netherlands, *De Volkskrant*, covered the study extensively, in an article called "NWO discriminates against female scientists". The study of Van der Lee and Ellemers, as well as the media coverage and public debate after the study, led NWO to re-evaluate their policy, and more actively strive for gender equality in research funding.

From that point onwards, several measures were implemented to counter gender inequality in research funding. First, panelists in the TP got implicit bias training. At first, these trainings were given face to face, whereas nowadays all panelists watch a video that addresses implicit bias before they start with their evaluations. Second, the wording of the grant calls and instructions for panelists and reviewers changed. Van der Lee and Ellemers found that the majority of the wording was masculine, which led them to argue that the policy goal of NWO (achieving gender inequality) was not reflected by the instructions that those involved in the grant evaluation. Since the study by Van der Lee and Ellemers, NWO explicitly mentions the proportion of female applicants and awardees, to show the extent to which gender inequality was obtained.

To summarize, we argue that during the 2010s, NWO increasingly emphasized gender equality in research funding. We therefore expect the probability of female applicants to obtain a grant in the TP to increase over the observed time period of this study (2012–2021).

## Data and methods

In our study, we define gender effects as differences between success rates of men and women that cannot be attributed to coincidence. Gender effects include both gender bias (i.e. the effects of (unconscious) prejudice against a gender) as well as any other effects that cause systematic deviations in performance of men and women in academia.

The goal of this study is to test whether observed gender differences in the success rate of the Talent Programme grants can be attributed to coincidence or not. More precisely, we consider the following research question: 'In absence of any gender effects in quality of applications and the considerations of the assessment committee, what is the probability of finding at least the same gender difference as was found in the data of 2012–2021?'. We will answer this research question using publicly available information on the number of applications and grants, by year, gender and research domain.

We have looked at all research grants from 2012 to the most recent grants at the moment of writing. We have included all data that were published on NWO's website until and including 26 July 2023. We only assessed the aggregated table with total number of applications and grants per gender, domain and year and did not study individal participants' data. We restricted our attention to the publicly available data: numbers of applications and numbers of funded projects, per gender and domain. Throughout this study, the calender year mentioned refers to the year of the funding decision, which usually is the year after the grant submission.

Here, we have focused on the period from 2012 onwards. The previous period, up to 2012, had already been assessed before [10]. Since NWO took its measures after publication of that paper (in 2015), the time period chosen (2012–now) allows us to investigate the possible effects of the new policy. All data discussed here have been obtained from NWO's website (see [25] for the Veni data, from where links to the data Vidi and Vici are available. All data are also provided at https://osf.io/8bfaz/).

For these programmes, NWO distinguishes five research fields:

- ENW: science

- SGW: social sciences and humanities

- TTW: applied and engineering sciences

- ZonMW: health research

- DO: cross-domain/interdisciplinary. (This domain has been cancelled as of 2020).

For each year and each field, we have recorded the number of submitted applications and granted applications for men and women separately. NWO publicly shares the necessary information for most but not all years, see the Supplementary Material for a detailed overview.

Starting in 2021, the system for the fields SGW, TTW and ZonMW has changed. Rather than giving all eligible academics the opportunity to submit a proposal and fund a proportion of these (as still is the case with ENW), these fields now work with pre-application. All eligible candidates can submit a short research idea and short cv and only a selection of them is invited to submit a full research proposal. As NWO does not publish the gender balance of submitted pre-applications, data for this round cannot be included in the analyses.

## The models

To model the probability of success, $p_i$, of a given application, we employ logistic regression (or binomial generalized linear models [26]). In these models, the expected logodds of $p_i$, log $(p_i/(1 − p_i))$ are predicted on the basis of a number of predictors. In our case, the success probabilities are predicted based on gender of the applicant, the field of study, and the year of application. Unfortunately we do not have information on the institutional affiliation of all applicants.

We distinguish four different models, of increasing complexity, based on these predictors:

1. Model 1: gender, field and year are used as additive predictors.

2. Model 2: as Model 1, but with an interaction between gender and year: the gender effect can differ per year.

3. Model 3: as Model 2, but with also an interaction between gender and field.

4. Model 4: as Model 3, but with also an interaction between year and field, i.e. all three second-order interactions.

Data for the three tiers are analyzed separately. Model fit and model parsimony are assessed through the Akaike Information Criterion.

The first model is specified by

$$\log\left(\frac{p_i}{1 − p_i}\right) = \beta_0 + \beta_M D_{M,i} + \beta_{DO} D_{DO,i} + \beta_{ENW} D_{ENW,i} + \beta_{TTW} D_{TTW,i} +$$
$$\beta_{ZonMw} D_{ZonMw,i} + \beta_{Year} \text{Year}_i + \varepsilon_i.$$

Here, $D_{X,i}$ is used as notation for the dummy variable (also known as the Kronecker delta $\delta_{X,i}$) indicating whether person $i$ belongs to class $X$ (then $D_{X,i} = 1$) or not (then $D_{X,i} = 0$). A class $X$ can stand for a research field, e.g. ENW or a gender ('M' is used as notation for male applicants, with female being the reference group for gender). The field SGW is chosen as reference field, as this field had the largest number of applications. (Note that the choice of reference fields is arbitrary: any other choice would have yielded exactly the same predicted success rates.) Variable 'Year' is included to measure the longitudinal effects. This variable is coded as 1 for 2012, 2 for 2013, . . ., 10 for 2021.

Subsequently, Model 2 is specified by

$$\log\left(\frac{p_i}{1-p_i}\right) = \beta_0 + \beta_M D_{M,i} + \beta_{DO} D_{DO,i} + \beta_{ENW} D_{ENW,i} + \beta_{TTW} D_{TTW,i} +$$
$$\beta_{ZonMw} D_{ZonMw,i} + \beta_{Year} \text{Year}_i + \beta_{M,Year} \times \text{Year}_i \times D_{M,i} + \varepsilon_i,$$

thus with an additional interaction term $\beta_{M,Year} \times \text{Year}_i \times D_{M,i}$. Analogously, in Model 3, interaction terms between gender and field are added, while Model 4 adds interaction terms for year and field to that.

All computations have been performed in *R* (version 4.1.2; [27]). The analyses of variance have been carried out using the *R* package 'car' [28].

## Results

The full dataset consists of a total of 16, 249 applications (6, 907 from female applicants, 9, 342 from male applicants). Out of these, 2, 449 have been granted (1,067 for female applicants, i.e. a success rate of 15.4%; and 1,382, for male applicants, i.e. a 14.8% success rate). There were no applicants that did not declare a gender, nor did any candidate declare a gender other than male or female. With 10, 076 applicants and 1, 472 funded applicants, the Veni tier is by far the largest tier. All descriptives are provided in Table 1. Note that in absolute numbers, male applicants outnumber female applicants and this gap grows with the tiers. In relative numbers, i.e. success rate, however, male applicants do not outperform female applicants, as discussed below.

As the first tier consists of 62% of all applications and 60% of all grants, we focus on this (Veni) scheme first, and in most detail. We find that all four models described predict lower success percentages for male applicants than for female applicants. Furthermore, clear differences in success rates between fields are observed, which is in line with previous studies on NWO's Veni grants [2, 12]. To avoid the Simpson's paradox fallacy [2, 12], all models take field of study into account.

Table 2 displays the results of an analysis of variance on the four models, and Table 3 displays the AIC-comparisons. The latter table clearly demonstrates that inclusion of a *gender × year* interaction is beneficial (Model 2). Model 3, which additionally includes the four *gender × field* interactions, has an even lower AIC-score, indicating that the gender gap changes over time for all fields. On the other hand, the addition of the *year × field* terms in Model 4 provides no significant improvement to the model fit ($p = .385$), as indicated by a higher AIC-value. Thus, we will look at Model 3 in more detail, as presented in Table 4. An explanation on how to interpret the coefficients of Table 4 is given in S1 Appendix. In S2 Appendix the R code of the analyses is provided, which is also available from https://osf.io/8bfaz/. This, in combination with the data will provide full results of the three other models.

Fig 1 represents the observed success probabilities and the predicted success probabilities according to Model 3 over the years considered. In this Figure, we present a graph for each

**Table 1. Numbers of applications and project fundings.**

| | Veni | | Vidi | | Vici | | Total | |
|---|---|---|---|---|---|---|---|---|
| | Applications | Granted | Applications | Granted | Applications | Granted | Applications | Granted |
| Women | 4,590 | 695 | 1,588 | 268 | 729 | 104 | 6,907 | 1,067 |
| Men | 5,486 | 777 | 2,400 | 411 | 1,456 | 196 | 9,342 | 1,382 |
| Total | 10,076 | 1472 | 3,988 | 679 | 2,185 | 300 | 16,249 | 2,449 |

**Table 2. Analysis of variance of the four models for the Veni data.** The $\chi^2$-values display the Wald test statistics, the other two columns per model the corresponding degrees of freedom and $p$-values.

| | Model 1 | | | Model 2 | | | Model 3 | | | Model 4 | | |
| --- | --- | --- | --- | --- | --- | --- | --- | --- | --- | --- | --- | --- |
| | df | $\chi^2$ | $p$-value | df | $\chi^2$ | $p$-value | df | $\chi^2$ | $p$-value | df | $\chi^2$ | $p$-value |
| Gender | 1 | 6.572 | .010 | 1 | 6.295 | .012 | 1 | 10.355 | .001 | 1 | 10.541 | .001 |
| Field | 4 | 80.256 | < .001 | 4 | 81.135 | < .001 | 1 | 73.333 | < .001 | 1 | 20.899 | < .001 |
| Year | 1 | .982 | .322 | 1 | 7.185 | .007 | 4 | 6.331 | .012 | 4 | 0.447 | .504 |
| Gender × Year | – | – | – | 1 | 21.166 | < .001 | 1 | 18.967 | < .001 | 1 | 20.142 | < .001 |
| Gender × Field | – | – | – | – | – | – | 4 | 16.591 | .002 | 4 | 14.965 | .005 |
| Year × Field | – | – | – | – | – | – | – | – | – | 4 | 4.173 | .383 |

field. In Fig 2 we aggregate the figures for the five domains into a single figure, using the numbers of applications per field as weights. All graphs in Figs 1 and 2 show a positive trend for grant succes rates for females and a (corresponding) negative one for males. The year at which the two lines cross varies per field. For DO, ENW and TTW the crossing takes place around 2012, where our dataset starts, whereas for SGW (around 2017) and ZonMw (around 2018), they happen later in time—although the uncertainty in these predictions is considerable. A crossing can also be observed in the aggregate predictions of Fig 2, roughly around the year 2015. As seen in Fig 1, there is considerable distance between certain observations and the corresponding predictions. This calls for some caution: whereas the model is sufficient to estimate the gender effect as a whole, it will not be sufficient for predictions for individual combinations of gender, year and field, let alone extrapolations to future (or past) years. Note that the

**Table 3. Comparison between the four models using the Akaike Information Criterion for the Veni data.**

| | df | AIC |
| --- | --- | --- |
| Model 1 | 7 | 478.04 |
| Model 2 | 8 | 458.77 |
| Model 3 | 12 | 450.24 |
| Model 4 | 16 | 454.05 |

**Table 4. Results for Model 3 for the Veni data.** Field SGW is the reference field, and Female is the reference gender. Note that $p$-values haven't been adjusted for multiple testing (a model for each of the three tiers) yet.

| | $\hat{\beta}$ | SE | $p$-value |
| --- | --- | --- | --- |
| Intercept | -2.164 | 0.096 | < .001 |
| Gender: Male | 0.417 | 0.130 | .001 |
| Year | 0.040 | 0.016 | 0.012 |
| Field: DO | 0.580 | 0.166 | < .001 |
| Field: ENW | 0.808 | 0.101 | < .001 |
| Field: TTW | 0.368 | 0.172 | 0.033 |
| Field: ZonMw | 0.046 | 0.123 | 0.712 |
| Gender: Male × Year | -0.094 | 0.022 | < .001 |
| Gender: Male × Field: DO | -0.470 | 0.256 | 0.067 |
| Gender: Male × Field: ENW | -0.391 | 0.137 | 0.004 |
| Gender: Male × Field: TTW | -0.426 | 0.219 | 0.052 |
| Gender: Male × Field: ZonMw | 0.170 | 0.175 | 0.331 |

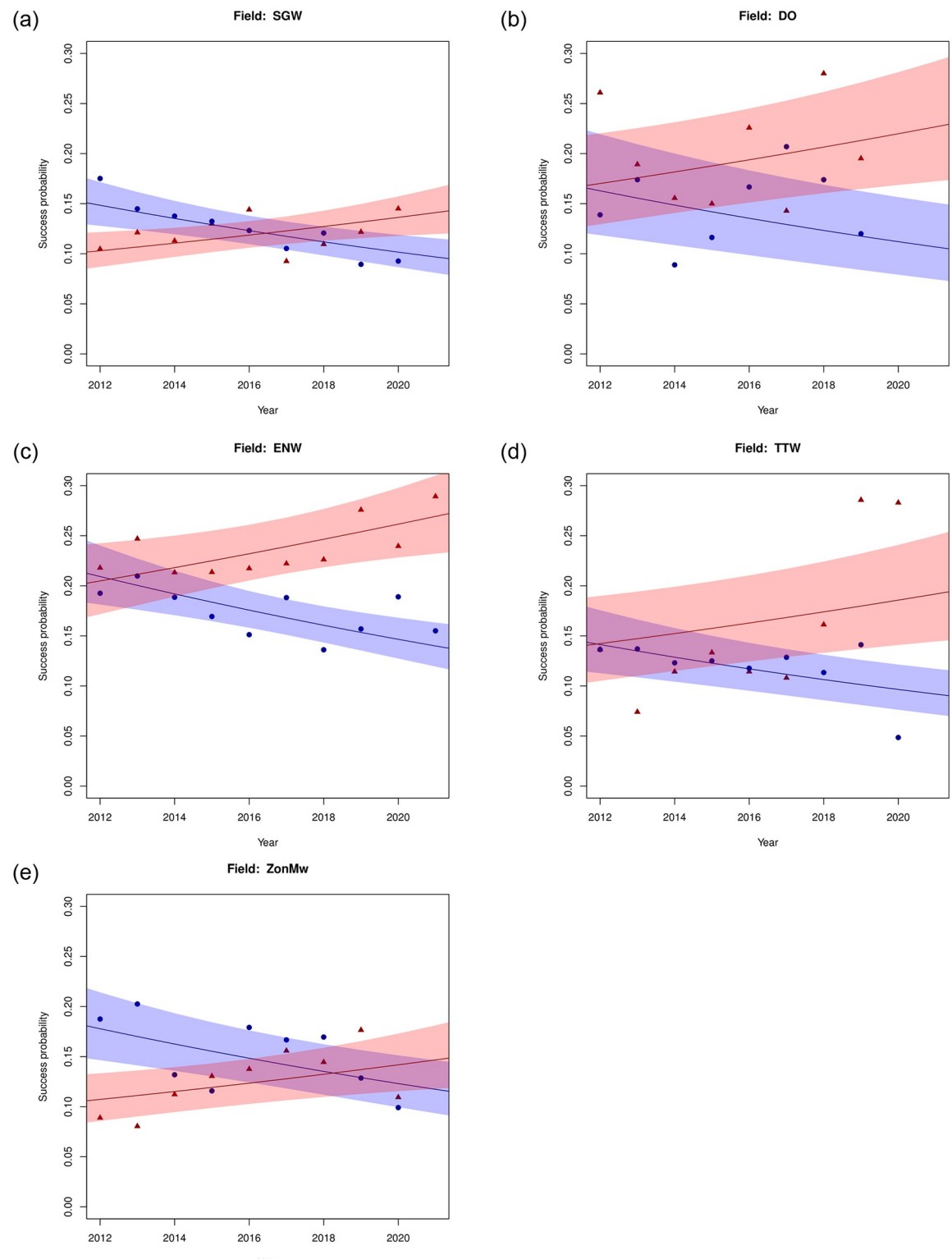

**Fig 1. Observed success probabilities (triangles for women, squares for men) and predictions according to Model 3 (increasing curves for women, decreasing curves for men) for the Veni data.** The shaded areas correspond to the 95% prediction intervals.

## All fields

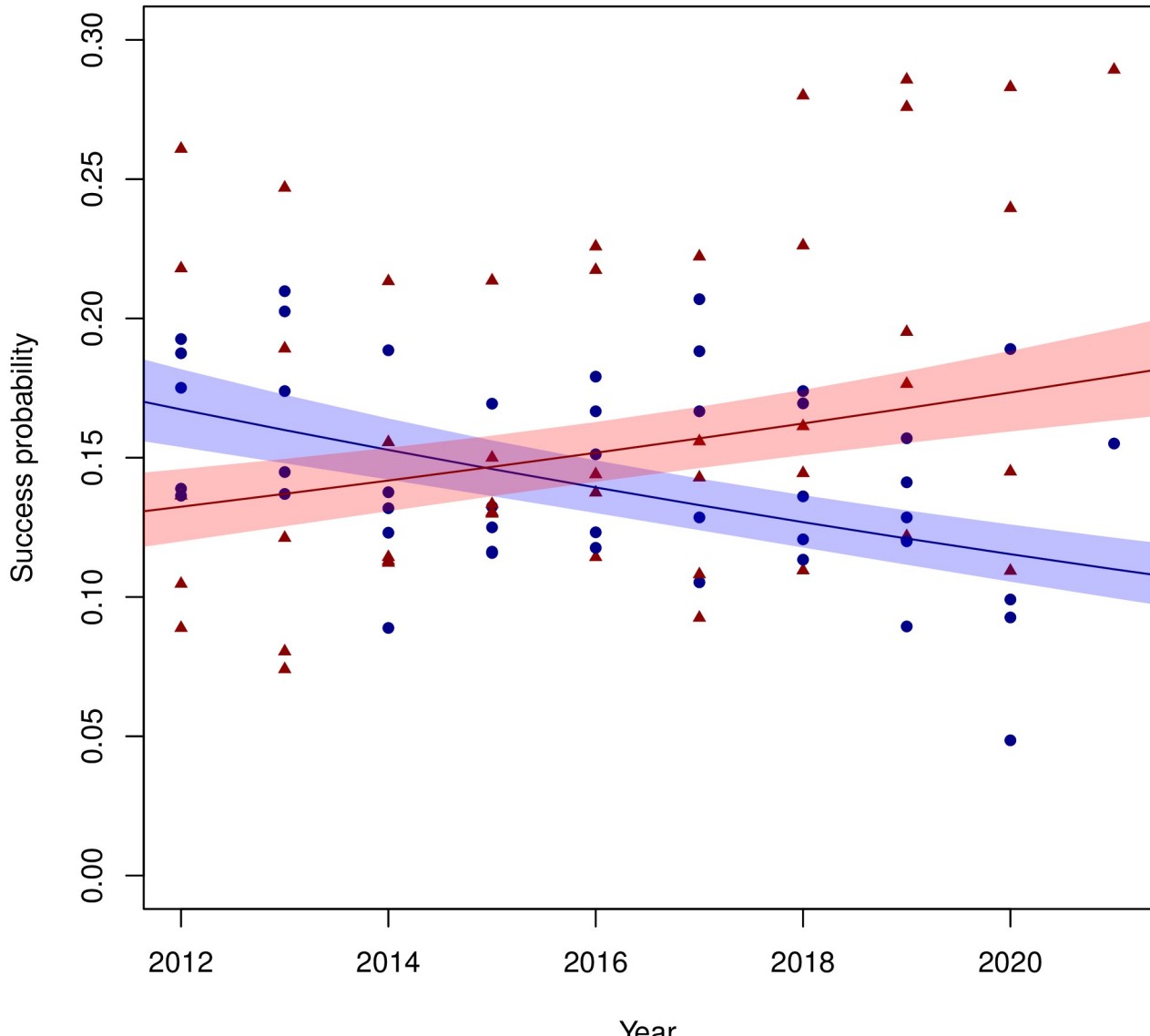

**Fig 2. Observed success probabilities and predictions according to Model 3 for the Veni data, aggregated over all five fields.** For an explanation of the symbols and colours, see the caption of Fig 1.

uncertainty in the moment of crossing is also considerable, making it difficult to assess when the success rate of female applicants overtakes those of male applicants precisely. Still, this does not diminish the significant change in gender effects over time.

In Tables 5 and 6 all predicted success probabilities for the Veni for all four models are listed.

An alternative way to present the results is through odds ratios (OR) rather than probabilities. The odds ratio compares predicted success probabilities for men with those for women. An OR >1 indicates a higher success probability for male applicants, OR<1 does so for

**Table 5. Predicted success probabilities, according to the four models [1/2].**

| Year | Field | Model 1 | | Model 2 | | Model 3 | | Model 4 | |
|------|-------|---------|------|---------|------|---------|------|---------|------|
| | | Women | Men | Women | Men | Women | Men | Women | Men |
| 2012 | SGW | 0.134 | 0.118 | 0.109 | 0.139 | 0.103 | 0.148 | 0.114 | 0.163 |
| 2013 | SGW | 0.133 | 0.117 | 0.113 | 0.132 | 0.107 | 0.142 | 0.115 | 0.152 |
| 2014 | SGW | 0.132 | 0.116 | 0.118 | 0.126 | 0.111 | 0.135 | 0.117 | 0.142 |
| 2015 | SGW | 0.130 | 0.115 | 0.122 | 0.120 | 0.115 | 0.129 | 0.118 | 0.132 |
| 2016 | SGW | 0.129 | 0.113 | 0.127 | 0.114 | 0.119 | 0.123 | 0.120 | 0.123 |
| 2017 | SGW | 0.128 | 0.112 | 0.131 | 0.108 | 0.123 | 0.117 | 0.121 | 0.114 |
| 2018 | SGW | 0.127 | 0.111 | 0.136 | 0.103 | 0.127 | 0.112 | 0.123 | 0.106 |
| 2019 | SGW | 0.126 | 0.110 | 0.141 | 0.098 | 0.132 | 0.107 | 0.124 | 0.098 |
| 2020 | SGW | 0.125 | 0.109 | 0.146 | 0.093 | 0.136 | 0.102 | 0.126 | 0.091 |
| 2021 | SGW | 0.123 | 0.108 | 0.152 | 0.089 | 0.141 | 0.097 | 0.127 | 0.084 |
| 2012 | DO | 0.183 | 0.162 | 0.151 | 0.189 | 0.170 | 0.163 | 0.159 | 0.156 |
| 2013 | DO | 0.181 | 0.160 | 0.156 | 0.181 | 0.176 | 0.156 | 0.168 | 0.151 |
| 2014 | DO | 0.180 | 0.159 | 0.162 | 0.172 | 0.182 | 0.149 | 0.176 | 0.146 |
| 2015 | DO | 0.178 | 0.157 | 0.168 | 0.165 | 0.188 | 0.142 | 0.185 | 0.142 |
| 2016 | DO | 0.176 | 0.156 | 0.173 | 0.157 | 0.194 | 0.135 | 0.195 | 0.137 |
| 2017 | DO | 0.175 | 0.154 | 0.180 | 0.150 | 0.200 | 0.129 | 0.204 | 0.133 |
| 2018 | DO | 0.173 | 0.153 | 0.186 | 0.143 | 0.206 | 0.123 | 0.214 | 0.129 |
| 2019 | DO | 0.172 | 0.152 | 0.192 | 0.136 | 0.213 | 0.118 | 0.225 | 0.125 |
| 2020 | DO | 0.170 | 0.150 | 0.199 | 0.129 | 0.220 | 0.112 | 0.236 | 0.121 |
| 2021 | DO | 0.169 | 0.149 | 0.206 | 0.123 | 0.227 | 0.107 | 0.247 | 0.117 |
| 2012 | ENW | 0.218 | 0.194 | 0.182 | 0.226 | 0.205 | 0.209 | 0.193 | 0.201 |
| 2013 | ENW | 0.217 | 0.192 | 0.188 | 0.216 | 0.211 | 0.200 | 0.202 | 0.195 |
| 2014 | ENW | 0.215 | 0.191 | 0.194 | 0.207 | 0.218 | 0.192 | 0.211 | 0.188 |
| 2015 | ENW | 0.213 | 0.189 | 0.201 | 0.198 | 0.225 | 0.184 | 0.220 | 0.181 |
| 2016 | ENW | 0.211 | 0.187 | 0.208 | 0.189 | 0.232 | 0.176 | 0.229 | 0.175 |
| 2017 | ENW | 0.209 | 0.186 | 0.215 | 0.180 | 0.239 | 0.168 | 0.239 | 0.169 |
| 2018 | ENW | 0.208 | 0.184 | 0.222 | 0.172 | 0.246 | 0.161 | 0.249 | 0.163 |
| 2019 | ENW | 0.206 | 0.183 | 0.229 | 0.164 | 0.254 | 0.153 | 0.260 | 0.157 |
| 2020 | ENW | 0.204 | 0.181 | 0.237 | 0.157 | 0.262 | 0.147 | 0.270 | 0.152 |
| 2021 | ENW | 0.203 | 0.180 | 0.245 | 0.150 | 0.269 | 0.140 | 0.281 | 0.146 |

female applicants and OR = 1 indicates no gender difference. Model 1 predicts the same OR for each year and field, which is *OR* = .862 (95% CI = [.770, .966] and is depicted by the dashed line in Fig 3. Model 2 introduces a temporal effect and the predicted OR are visualised with the black curve in Fig 3. Model 3 introduces field differences and the five resulting curves are shown in this figure as well. As Model 4 is no improvement over Model 3, this model is not included in the visualisation. Table 7 lists the OR for all four models in a similar style as Tables 5 and 6.

In the same vein as the analyses for the first tier, the Vidi and Vici tiers are analysed. Unlike in the Veni data, for both these tiers the best performing model is Model 1, the model without any interactions of gender with one of the other variables (Table 8). Furthermore, neither in the Vidi nor in the Vici data a significant effect of gender is found (Table 9). Thus, in contrast with the Veni data, there is no evidence for any gender effect in success rate: no base rate difference, nor a change of this effect over time. The lack of significant gender effects is illustrated in Fig 4.

**Table 6. Predicted success probabilities, according to the four models [2/2].**

|  |  | Model 1 | | Model 2 | | Model 3 | | Model 4 | |
|---|---|---|---|---|---|---|---|---|---|
| Year | Field | Women | Men | Women | Men | Women | Men | Women | Men |
| 2012 | TTW | 0.148 | 0.131 | 0.121 | 0.153 | 0.142 | 0.141 | 0.119 | 0.123 |
| 2013 | TTW | 0.147 | 0.129 | 0.126 | 0.146 | 0.147 | 0.135 | 0.128 | 0.121 |
| 2014 | TTW | 0.146 | 0.128 | 0.130 | 0.139 | 0.152 | 0.129 | 0.138 | 0.120 |
| 2015 | TTW | 0.144 | 0.127 | 0.135 | 0.133 | 0.158 | 0.123 | 0.148 | 0.118 |
| 2016 | TTW | 0.143 | 0.126 | 0.140 | 0.126 | 0.163 | 0.117 | 0.158 | 0.116 |
| 2017 | TTW | 0.142 | 0.125 | 0.145 | 0.120 | 0.168 | 0.112 | 0.170 | 0.115 |
| 2018 | TTW | 0.140 | 0.123 | 0.151 | 0.114 | 0.174 | 0.106 | 0.182 | 0.113 |
| 2019 | TTW | 0.139 | 0.122 | 0.156 | 0.109 | 0.180 | 0.101 | 0.194 | 0.112 |
| 2020 | TTW | 0.138 | 0.121 | 0.162 | 0.104 | 0.186 | 0.097 | 0.207 | 0.110 |
| 2021 | TTW | 0.137 | 0.120 | 0.167 | 0.098 | 0.192 | 0.092 | 0.221 | 0.109 |
| 2012 | ZonMw | 0.150 | 0.132 | 0.122 | 0.155 | 0.107 | 0.178 | 0.105 | 0.176 |
| 2013 | ZonMw | 0.148 | 0.130 | 0.127 | 0.147 | 0.111 | 0.170 | 0.109 | 0.169 |
| 2014 | ZonMw | 0.147 | 0.129 | 0.132 | 0.141 | 0.115 | 0.163 | 0.114 | 0.162 |
| 2015 | ZonMw | 0.146 | 0.128 | 0.136 | 0.134 | 0.119 | 0.155 | 0.118 | 0.155 |
| 2016 | ZonMw | 0.144 | 0.127 | 0.141 | 0.128 | 0.123 | 0.148 | 0.123 | 0.148 |
| 2017 | ZonMw | 0.143 | 0.126 | 0.147 | 0.121 | 0.128 | 0.142 | 0.128 | 0.142 |
| 2018 | ZonMw | 0.142 | 0.125 | 0.152 | 0.116 | 0.132 | 0.135 | 0.134 | 0.136 |
| 2019 | ZonMw | 0.140 | 0.123 | 0.158 | 0.110 | 0.137 | 0.129 | 0.139 | 0.130 |
| 2020 | ZonMw | 0.139 | 0.122 | 0.163 | 0.105 | 0.142 | 0.123 | 0.145 | 0.124 |
| 2021 | ZonMw | 0.138 | 0.121 | 0.169 | 0.099 | 0.147 | 0.117 | 0.150 | 0.119 |

## Discussion

Our results point to two key findings. First, we find that for the Veni tier, female applicants are more likely to be awarded with a grant than male applicants. In absence of any gender effects in the quality of applications and considerations of the assessment committee, the probability of finding at least the gender difference was smaller than 0.001. Thus, the null hypothesis of no gender differences is rejected. Interpretation of the results (e.g. Figs 1 and 2 and the odds ratios in Tables 5 and 6) reveals that this gender difference is to the favour of women. For the other two tiers, Vidi and Vici, no significant gender effects were found.

Second, we find that this probability changes over time. In the Veni tier, across all fields, we find that the success probability of female applicants increases over time, at the cost of that of male applicants. Gender effects in the Veni tier have shifted over the years, in favour of females. It could hence be that the measures taken by NWO to combat gender effects against women—introduced after the Veni study [10]—have indeed been successful. However, since differences in success rate per gender in the Veni's were small, or possibly even absent, to start with (see [2, 10, 12] and Figs 1 and 2), they have increased the funding probability of female applicants relative to that of male applicants.

We have to be cautious with this intepretation, however. We are not able to make a statement about men being disadvantaged. Obtaining funding is conditional on applying to a grant, and it is very likely that there are gender differences here too—and that these also shifted over time. Furthermore, our model is correlational and not causal. We also lack the data on the number of applicants that received extensions due to child birth and child care and thus cannot measure the extent to which this affected the results. The purpose of this paper is not to find the mechanisms behind observed gender effects, nor to state whether or not they are due

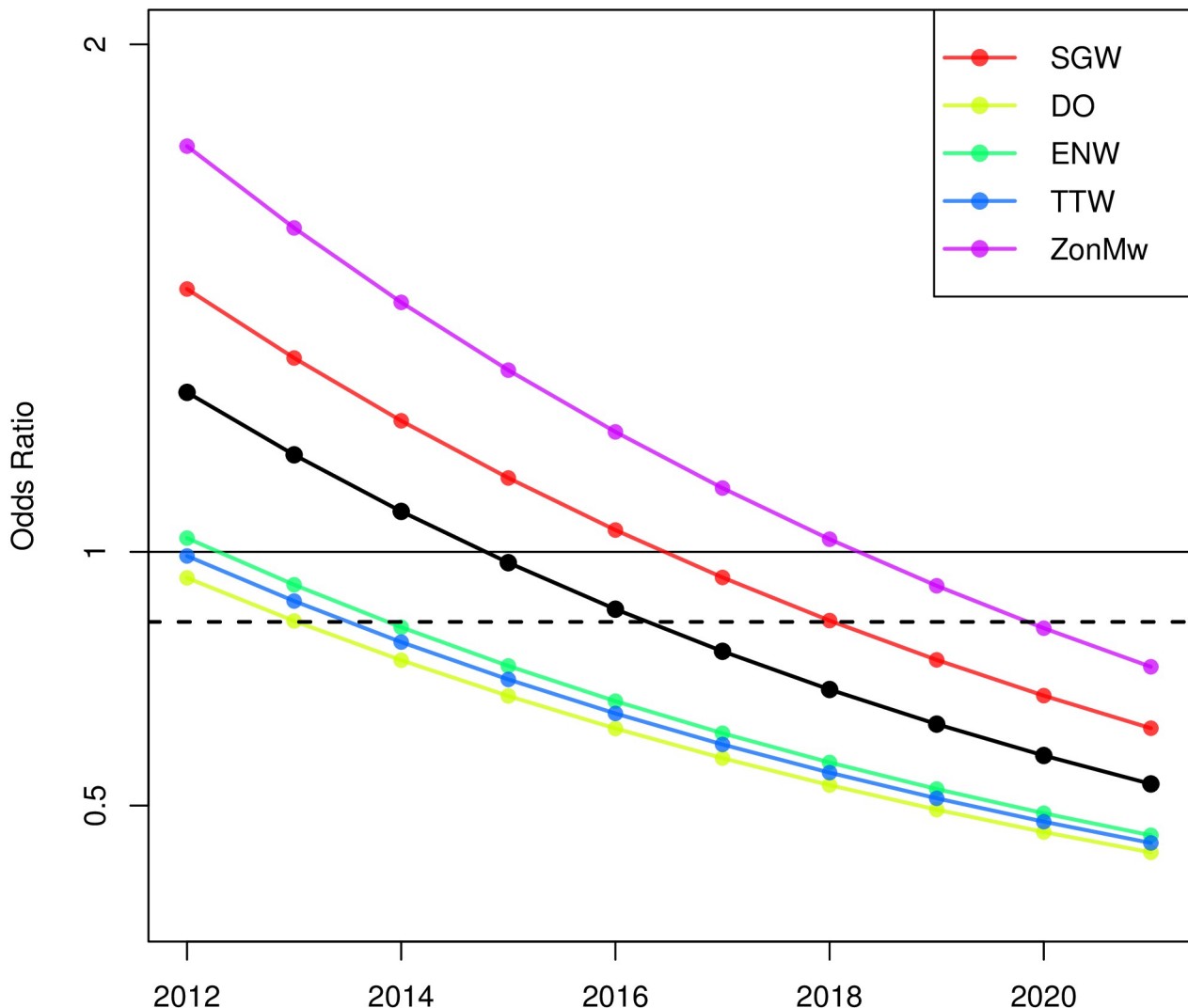

**Fig 3. Odds ratios (OR) for the different combinations of year and field, for Models 1, 2 and 3.** OR>1 indicates a higher success probability for men. The OR for Model 1 is depicted by the dashed line, the OR for Model 2 by the solid black line, the OR for Model 3 by the coloured lines.

to gender bias, but merely to answer the question whether the observed gender effects are statistically significant. They are in the Veni data. They are not in the other tiers.

Despite their relatively high success rates in the Veni scheme, however, it does appear that more women than men leave academia before reaching the second and third tier of the Talent Programme. The fact that the percentage of female applicants clearly declines over the tiers (46% for Veni, 40% for Vidi, 33% for Vici) supports this. This is in line with existing studies that argue that for women, careers in academia can be defined as a "leaky pipeline", where women leave academia at every career step [29].

A recent study [4] studied all Talent Programme data, including (confidential) scores from reviewers. These authors found that male applicants receive better reviewer scores than female applicants—indicative of gender effects in assessment. Yet, they also find evidence that external review scores were corrected for by the panels, mostly in the rebuttal phase. Furthermore, women are overrepresented at ranking positions just above the funding threshold. Combining

**Table 7. Odds ratios for the four different models.**

| Year | Field | Model 1 | Model 2 | Model 3 | Model 4 | Field | Model 1 | Model 2 | Model 3 | Model 4 |
|------|-------|---------|---------|---------|---------|-------|---------|---------|---------|---------|
| 2012 | SGW | 0.862 | 1.314 | 1.518 | 1.516 | TTW | 0.862 | 1.314 | 0.992 | 1.034 |
| 2013 | SGW | 0.862 | 1.191 | 1.382 | 1.375 | TTW | 0.862 | 1.191 | 0.903 | 0.938 |
| 2014 | SGW | 0.862 | 1.08 | 1.258 | 1.247 | TTW | 0.862 | 1.08 | 0.822 | 0.851 |
| 2015 | SGW | 0.862 | 0.979 | 1.146 | 1.132 | TTW | 0.862 | 0.979 | 0.749 | 0.772 |
| 2016 | SGW | 0.862 | 0.887 | 1.043 | 1.026 | TTW | 0.862 | 0.887 | 0.682 | 0.7 |
| 2017 | SGW | 0.862 | 0.804 | 0.95 | 0.931 | TTW | 0.862 | 0.804 | 0.62 | 0.635 |
| 2018 | SGW | 0.862 | 0.729 | 0.865 | 0.845 | TTW | 0.862 | 0.729 | 0.565 | 0.576 |
| 2019 | SGW | 0.862 | 0.661 | 0.787 | 0.766 | TTW | 0.862 | 0.661 | 0.514 | 0.523 |
| 2020 | SGW | 0.862 | 0.599 | 0.717 | 0.695 | TTW | 0.862 | 0.599 | 0.468 | 0.474 |
| 2021 | SGW | 0.862 | 0.543 | 0.652 | 0.631 | TTW | 0.862 | 0.543 | 0.426 | 0.43 |
| 2012 | DO | 0.862 | 1.314 | 0.949 | 0.972 | ZonMw | 0.862 | 1.314 | 1.799 | 1.831 |
| 2013 | DO | 0.862 | 1.191 | 0.864 | 0.882 | ZonMw | 0.862 | 1.191 | 1.638 | 1.661 |
| 2014 | DO | 0.862 | 1.08 | 0.786 | 0.8 | ZonMw | 0.862 | 1.08 | 1.492 | 1.507 |
| 2015 | DO | 0.862 | 0.979 | 0.716 | 0.726 | ZonMw | 0.862 | 0.979 | 1.358 | 1.367 |
| 2016 | DO | 0.862 | 0.887 | 0.652 | 0.659 | ZonMw | 0.862 | 0.887 | 1.236 | 1.24 |
| 2017 | DO | 0.862 | 0.804 | 0.594 | 0.597 | ZonMw | 0.862 | 0.804 | 1.126 | 1.125 |
| 2018 | DO | 0.862 | 0.729 | 0.54 | 0.542 | ZonMw | 0.862 | 0.729 | 1.025 | 1.02 |
| 2019 | DO | 0.862 | 0.661 | 0.492 | 0.492 | ZonMw | 0.862 | 0.661 | 0.933 | 0.926 |
| 2020 | DO | 0.862 | 0.599 | 0.448 | 0.446 | ZonMw | 0.862 | 0.599 | 0.85 | 0.84 |
| 2021 | DO | 0.862 | 0.543 | 0.408 | 0.405 | ZonMw | 0.862 | 0.543 | 0.773 | 0.762 |
| 2012 | ENW | 0.862 | 1.314 | 1.027 | 1.052 | | | | | |
| 2013 | ENW | 0.862 | 1.191 | 0.935 | 0.955 | | | | | |
| 2014 | ENW | 0.862 | 1.08 | 0.851 | 0.866 | | | | | |
| 2015 | ENW | 0.862 | 0.979 | 0.775 | 0.785 | | | | | |
| 2016 | ENW | 0.862 | 0.887 | 0.706 | 0.713 | | | | | |
| 2017 | ENW | 0.862 | 0.804 | 0.643 | 0.646 | | | | | |
| 2018 | ENW | 0.862 | 0.729 | 0.585 | 0.586 | | | | | |
| 2019 | ENW | 0.862 | 0.661 | 0.533 | 0.532 | | | | | |
| 2020 | ENW | 0.862 | 0.599 | 0.485 | 0.483 | | | | | |
| 2021 | ENW | 0.862 | 0.543 | 0.442 | 0.438 | | | | | |

**Table 8. Comparison between the four models using the Akaike Information Criterion for the Vidi and Vici data.**

| | df | AIC Vidi | AIC Vici |
|---------|-----|----------|----------|
| Model 1 | 7 | 339.10 | 276.71 |
| Model 2 | 8 | 340.67 | 278.28 |
| Model 3 | 12 | 347.03 | 284.81 |
| Model 4 | 16 | 351.87 | 283.86 |

**Table 9. ANOVA tables for the Vidi and Vici data.** The $p$-values haven't been adjusted for triple multiple testing.

| | Vidi | | | Vici | | |
|--------|-----|--------|---------|-----|--------|---------|
| | df | $\chi^2$ | $p$-value | df | $\chi^2$ | $p$-value |
| Gender | 1 | .500 | .480 | 1 | 1.144 | .285 |
| Field | 4 | 44.660 | < .001 | 4 | 9.000 | .061 |
| Year | 1 | .141 | .708 | 1 | 4.463 | .035 |

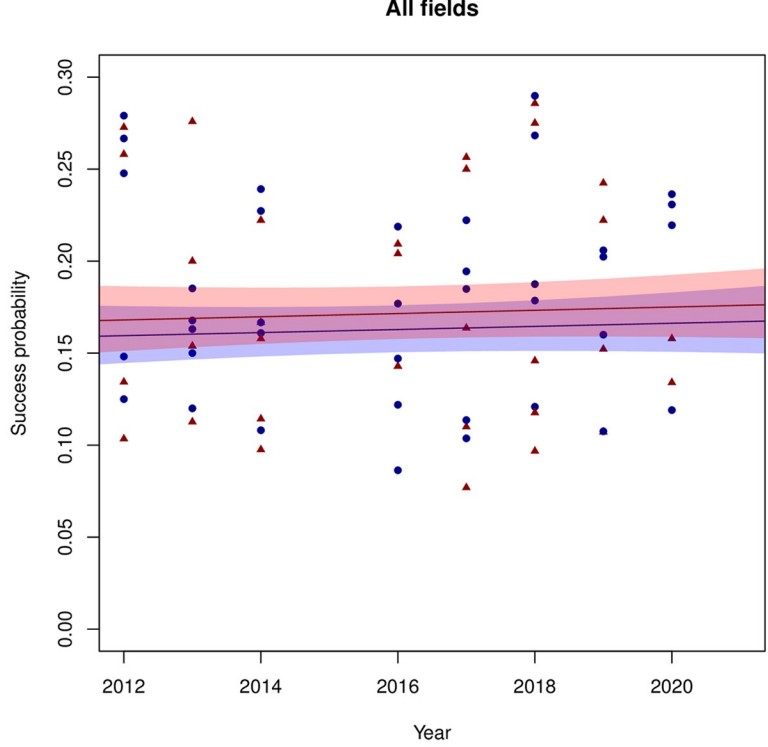

(b)

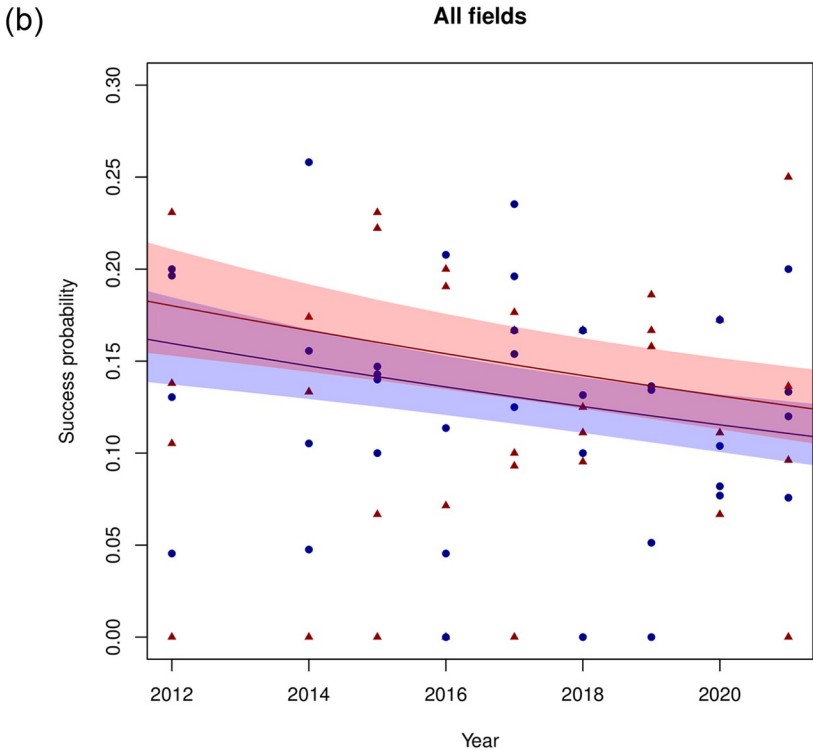

**Fig 4. Observed success probabilities and predictions according to Model 1 for the Vidi (left panel) and Vici (right panel) data, aggregated over the five domains.** For an explanation of the symbols and colours, see the caption of Fig 1.

the conclusions by [4] with our results, we hypothesize that the corrections performed by the juries may have gotten stronger over the years, yielding an overcorrection in recent times.

Our study also has several limitations. The biggest limitation is that we can only observe gender inequality in those who have applied for the Talent Program. Given that women decreasingly apply as the tiers get higher (relatively many more female applicants for the Veni than the Vici), it is likely that there is selection in who keep applying for the grants. In other words, it is likely that the average female applicant for the Vici is of higher quality than the average male applicant, given the gendered attrition over the schemes. Future research should investigate this question, using data not just of realized applications, but also investigating the pool of potential applicants.

More generally, our study provokes the question what policy funding agencies should do to prevent statistically relevant biases in the future. Clearly, to guarantee a proper feedback mechanism, a continuous, critical assessment of the available data over time is essential.

## Supporting information

**S1 Appendix. Interpretation of the logistic regression coefficients.**
(PDF)

**S2 Appendix. Analysis code.**
(PDF)

## Acknowledgments

The authors thank Mara Yerkes for helpful feedback on the manuscript.

## Author Contributions

**Conceptualization:** Casper Albers, Sense Jan van der Molen.

**Formal analysis:** Casper Albers.

**Investigation:** Sense Jan van der Molen.

**Methodology:** Casper Albers.

**Visualization:** Casper Albers.

**Writing – original draft:** Casper Albers, Sense Jan van der Molen.

**Writing – review & editing:** Casper Albers, Sense Jan van der Molen, Thijs Bol.

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
