## [Decision Letter · Decision Letter 0]

27 Oct 2023

PONE-D-23-27406Gender differences in Dutch research funding over time: A statistical investigation of the Innovation Scheme 2012–2021PLOS ONE

Dear Dr. Albers,

Thank you for submitting your manuscript to PLOS ONE. After careful consideration, we feel that it has merit but does not fully meet PLOS ONE’s publication criteria as it currently stands. Therefore, we invite you to submit a revised version of the manuscript that addresses the points raised during the review process.

We look forward to receiving your revised manuscript.

Kind regards,

Julian D. Cortes

Academic Editor

PLOS ONE

Journal Requirements:

3. Please upload a copy of Supporting Information Figure/Table/etc. which you refer to in your text on page 7,8 and 10.

**Additional Editor Comments:**

Dear author/s, thanks for submitting your work to PLoS ONE
, I contrasted the core sections of your work with our seven criteria for publication. In line/Besides reviewers' suggestions, consider the following recommendations:  1
Experiments, statistics, and other analyses are performed to a high technical standard and are described in sufficient detail.Extend your argument on the potential relevance of institutional affiliation in the in the probability of success in your model (in “Model” or “Discussion” sections?)Following good practices standards in statistical practice (https://www.nature.com/articles/s41562-021-01211-8) an additional section aimed to visualizing the data, besides the “Observed success probabilities” figures, could be added to excel the descriptive/explicative interpretation of results in logistic regression models, such as: -
Correlation Matrix-
ROC Curve-
Confusion Matrix2
Conclusions are presented in an appropriate fashion and are supported by the data.Extend your conclusion section with: -
The paragraph in which it is stated that “more women than men leave academia before” is of great interest for the community of practice. Extend your conclusions on that. -
The limitations of the study. -
Further research agenda derivates from it. 3
The article is presented in an intelligible fashion and is written in standard English.Extend the abstract content with the following criteria: “Background or Introduction – What is currently known? Start with a brief, 2 or 3 sentence, introduction to the research area. Objectives or Aims – What is the study and why did you do it? Clearly state the research question you’re trying to answer. Methods – What did you do? Explain what you did and how you did it. Include important information about your methods, but avoid the low-level specifics. Some disciplines have specific requirements for abstract methods. CONSORT for randomized trials. STROBE for observational studies PRISMA for systematic reviews and meta-analyses. Results – What did you find? Briefly give the key findings of your study. Include key numeric data (including confidence intervals or p values), where possible. Conclusions – What did you conclude? Tell the reader why your findings matter, and what this could mean for the ‘bigger picture’ of this area of research.”4
The article adheres to appropriate reporting guidelines and community standards for data availability.Data and code for replication was submitted by the authors: https://osf.io/8bfaz/. Consider mention it in the manuscript body (“Data & materials” section?).  I hope you can incorporate the above suggestions to improve your already valuable work. Sincerely, Julián D. Cortés Associated Editor

Reviewers' comments:

Reviewer's Responses to Questions

**Comments to the Author**

1. Is the manuscript technically sound, and do the data support the conclusions?

Reviewer #1: Partly

Reviewer #2: Yes

2. Has the statistical analysis been performed appropriately and rigorously? 

Reviewer #1: No

Reviewer #2: Yes

3. Have the authors made all data underlying the findings in their manuscript fully available?

Reviewer #1: Yes

Reviewer #2: Yes

4. Is the manuscript presented in an intelligible fashion and written in standard English?

Reviewer #1: Yes

Reviewer #2: Yes

5. Review Comments to the Author

Reviewer #1: As regards to form, the paper is fit for publication, and I have no comments whatsoever in that regard. However, I believe that two relatively important issues prevent it from being published in its actual form.

The first one, and probably the most important, relates to results interpretation. In the discussion section, the authors say: "In absence of any gender effects in the quality of applications and considerations of the assessment committee, the probability of finding at least the gender difference was smaller than 0.001, i.e. there is a very significant gender difference to the favour of women." While the first part of this sentence is adequate, the last part is plain wrong. P-values in logistic regression are to be interpreted in the same way as in any other frequentist statistical test, that is, as the probability of obtaining results at least extreme under the hypothesis we aim to nullify. If the probability is lower than a specified threshold, we can safely reject that hypothesis, and that's it. P-values do not say anything about the significance or the plausibility of any other hypothesis, since the hypothesis we aim to nullify is the only one being tested. In order to say anything more, you need to measure effect size. We invite the authors to look at how this can be done as regards to logistic regression; as a suggestion, a thorough analysis of odds ratios should be appropriate.

The second point relates to a comment made at the beginning of the article, where the authors mention that "in certain situations, such as childcare responsibilities, terms [for the different funding programs] can be extended". Given the pervasive an systemic gender imbalance all over the world as regards to childcare, it is reasonable to hypothesize that this childcare extension could have been a major factor in improving the gender imbalance situation as regards to the Veni grants, especially since it is probably during that period the necessities of child caring places the greatest strain on female scholars' careers. We suppose that data regarding such extensions is not available as of now; if it were, I don't think this research would be complete or publishable without a thorough analysis of grant extensions. But the unavailability of such data doesn't make the case of childcare extensions for Veni grants less crucial to the general study of gender imbalances in research funding. In light of this, we believe it is imperative that the authors discuss this matter thoroughly, at the very least as an unfortunate and (perhaps) unavoidable limitation of their research.

Reviewer #2: The manuscript aims at analyzing gender differences in award rates for research grants of the Dutch Research Council (NWO; Veni, Vidi, Vici). In particular, the focus is on temporal developments: Do gender inequalities have changed over time? To do so, the authors use publicly available NWO data on submissions and awarded grants (by gender) for the years 2012 to 2021. The analyses are mainly based on logistic regressions with interaction terms to detect temporal changes of the gender effect. The key finding is that throughout the observation period the success probabilities of female applicants have increased compared to success rates of male applicants.

The authors convincingly show how their work adds to the literature. I think that the research question can be answered well with the data. The analyzes are well comprehensible and well documented so that replication is possible. Although the authors mainly intend to provide descriptions of developments, they also shortly discuss possible explanations for the observed results. The discussion of their results reveals that they are aware of the limitations of their study. I really appreciate the clear writing style.

A few more detailed comments:

Lines 66-68: I would not say that all models indicate lower success rates of males. For example, model 3 shows a positive gender (main) effect, meaning that at year=0 males have an advantage over females. By the way, I would recommend to code the variable “year” slightly different, so that the value zero refers to the first year in the data (i.e., 2012=0, … , 2021=9)

Line 77: As far as I know Lutter & Schröder do find evidence for gender inequalities controlling for publication output (in favor of female researchers).

6. PLOS authors have the option to publish the peer review history of their article (what does this mean?). If published, this will include your full peer review and any attached files.

Reviewer #1: No

Reviewer #2: No

---

## [Author Response · Author response to Decision Letter 0]

1 Dec 2023

Please see the uploaded PDF file for all responses to comments of the editor and reviewers

---

## [Decision Letter · Decision Letter 1]

3 Jan 2024

Gender differences in Dutch research funding over time: A statistical investigation of the Innovation Scheme 2012–2021

PONE-D-23-27406R1

Dear Dr. Albers,

We’re pleased to inform you that your manuscript has been judged scientifically suitable for publication and will be formally accepted for publication once it meets all outstanding technical requirements.

Kind regards,

Stijn Michielse, PhD

Academic Editor

PLOS ONE

Additional Editor Comments (optional):

Thanks for the revisions. Two reviewers had reviewed the manuscript and came independently to the conclusion that the manuscript is acceptable for publication.

Reviewers' comments:

Reviewer's Responses to Questions

**Comments to the Author**

1. If the authors have adequately addressed your comments raised in a previous round of review and you feel that this manuscript is now acceptable for publication, you may indicate that here to bypass the “Comments to the Author” section, enter your conflict of interest statement in the “Confidential to Editor” section, and submit your "Accept" recommendation.

Reviewer #1: All comments have been addressed

Reviewer #2: All comments have been addressed

2. Is the manuscript technically sound, and do the data support the conclusions?

Reviewer #1: Yes

Reviewer #2: Yes

3. Has the statistical analysis been performed appropriately and rigorously? 

Reviewer #1: Yes

Reviewer #2: Yes

4. Have the authors made all data underlying the findings in their manuscript fully available?

Reviewer #1: Yes

Reviewer #2: Yes

5. Is the manuscript presented in an intelligible fashion and written in standard English?

Reviewer #1: Yes

Reviewer #2: Yes

6. Review Comments to the Author

Reviewer #1: (No Response)

Reviewer #2: Thank you for the revision. I think the manuscript has benefited from the additions. In my view, it is now acceptable for publication.

However, please correct the following points before publication:

Line 275: "the black curve in 3"  "the black curve in Figure 3"

Line 317: Include a reference at the end of the sentence

7. PLOS authors have the option to publish the peer review history of their article (what does this mean?). If published, this will include your full peer review and any attached files.

Reviewer #1: No

Reviewer #2: No

---

## [Editor Report · Acceptance letter]

9 Feb 2024

PONE-D-23-27406R1 

PLOS ONE

Dear Dr. Albers, 

I'm pleased to inform you that your manuscript has been deemed suitable for publication in PLOS ONE. Congratulations! Your manuscript is now being handed over to our production team.

Kind regards, 

on behalf of

Dr. Stijn Michielse 

Academic Editor

PLOS ONE